# Association between Serum Triglycerides and Prostate Specific Antigen (PSA) among U.S. Males: National Health and Nutrition Examination Survey (NHANES), 2003–2010

**DOI:** 10.3390/nu14071325

**Published:** 2022-03-22

**Authors:** Chengcheng Wei, Liang Tian, Bo Jia, Miao Wang, Ming Xiong, Bo Hu, Changqi Deng, Yaxin Hou, Teng Hou, Xiong Yang, Zhaohui Chen

**Affiliations:** 1Department of Urology, Union Hospital, Tongji Medical College, Huazhong University of Science and Technology, Wuhan 430074, China; chengchengwei@hust.edu.cn (C.W.); m202175937@hust.edu.cn (M.W.); xiong_ming@hust.edu.cn (M.X.); m202175919@hust.edu.cn (C.D.); m201975738@hust.edu.cn (Y.H.); h1aiyan@hust.edu.cn (T.H.); yangxiong1368@hust.edu.cn (X.Y.); 2Department of Urology, Wuhan Red Cross Hospital, Wuhan 430015, China; tianliang1116@163.com; 3People’s Hospital of Dongxihu District, Wuhan 430040, China; jiabo@xhdxhrmyy.wecom.work (B.J.); hubo@dxhormyy.wecom.work (B.H.)

**Keywords:** prostate-specific antigen, triglycerides, National Health and Nutrition Examination Survey (NHANES), prostate cancer, machine learning

## Abstract

(1) Background: Increasing evidence indicates that lipid metabolism may influence the concentration of prostate-specific antigen (PSA). However, the association between triglycerides and PSA remains unclear and complicated. Hence, we evaluated the correlation between triglycerides and PSA based on the U.S. National Health and Nutrition Examination Survey (NHANES) database. (2) Methods: A total of 2910 participants out of 41,156 participants fit into our study after conducting the screening from the 2003 to 2010 NHANES survey. Serum triglycerides were the independent variable of our study, and PSA was the dependent variable; (3) Results: In our study, the average age of chosen participants was 59.7 years (±12.7). After adjusting for covariates, the result indicated that for each additional unit of serum triglyceride (mg/dL), the PSA concentrations were reduced by 0.0043 ng/mL (−0.0082, −0.0005) with a statistical difference. Furthermore, we used machine learning of the XGBoost model to determine the relative importance of selected variables as well as constructed a smooth curve based on the fully adjusted model to investigate the possible linear relationship between the triglyceride and PSA concentrations. (4) Conclusions: The serum triglyceride is independently and negatively correlated with PSA among American males, which may make it hard to detect asymptomatic prostate cancer and diagnose at an advance stage with higher triglycerides due to detection bias.

## 1. Introduction

In 2020, prostate cancer (PCa) was recorded as the most common cancer diagnosed in men, accounting for 26%, and was the second cause of the estimated death of 34,130 cases in males [1]. Detection of serum prostate-specific antigen (PSA) concentrations plays an essential part in screening prostate cancer at an early age amenable to curative treatment [2]. Widespread use of PSA concentration assays has significantly reduced the overall disease-specific mortality and improved the detection rate of asymptomatic, which highly differentiated prostate cancer [3,4]. Nevertheless, numerous studies have shown that PSA concentrations could be affected by various other factors, which may occur as detection bias in prostate cancer [5,6,7]. Inopportune and unnecessary treatment may take place due to overdiagnosis or missed diagnosis influenced by the various factors [8,9]. The U.S. Preventive Services Task Force (USPSTF) recently updated their recommendation, which changed from a grade D of guidance against PSA as based screening to a grade C of advocating for an individual screening [2,3,10]. Screening for prostate cancer by detecting PSA remains highly controversial because of the limitations of random trials, insufficient clinical evidence, and numerous factors affecting PSA levels [9,11]. Missing PSA data, hard to monitor, and the variety of decisions on patients between different areas may also contribute to practice variations [12].

An increasing number of studies strongly indicate that lipid and glucose metabolism play a critical role in prostate cancer (PCa) development [13]. Some studies have shown that cholesterol plays a possible role in prostate cancer development, aggressiveness, and progression through the effects of inflammation and steroidogenesis [14,15]. Some studies have also reported that higher serum low-density lipoprotein-cholesterol (LDL-C) levels correlated with a higher risk of prostate cancer. Meanwhile, higher levels of serum high-density lipoprotein-cholesterol (HDL-C) were corrected with the nonaggressive disease and lower risk of overall prostate cancer [16,17,18]. However, conflicting explanations exist on the function of triglycerides in the process of prostate cancer. Some studies have illustrated that triglycerides promote the malignant progression of prostate cancer [19]. However, some studies have considered triglycerides as a protection factor because of the higher triglyceride detection in a normal population [20,21]. In fact, the association may occur between triglyceride PSA metabolism, which could cause the detection bias in prostate cancer diagnosis. Moreover, we found this phenomenon has not been previously reported.

We hypothesized that serum triglycerides may influence the PSA concentrations among men without diagnosis of prostate cancer in the U.S. population. The presence of possible association could create a detection bias in observational studies of the exposures and prostate cancer. To test the hypothesis, we investigated the U.S. National Health and Nutrition Examination Survey (NHANES) for secondary data analysis. After controlling a vast array of influencing factors, we attempted to clarify the association between serum triglycerides and PSA concentrations among U.S. men without prostate cancer as well as try to explain the conflicting results

## 2. Materials and Methods

### 2.1. Data Availability

Since 1960, the National Health and Nutrition Examination Survey (NHANES) has been designed to estimate the health and nutritional status of adults and children in the United States, which is conducted by the National Centers for Disease Control (CDC) and Prevention National Health Statistics Center. Population and methodological details are available at the NHANES website (www.cdc.gov/nchs/nhanes, accessed on 7 October 2021). The National Center approved NHANES protocols for the Health Statistics research ethics review board.

### 2.2. Study Population

Four cycles’ two-year data of the NHANES survey from 2003 to 2010 were integrated into our study. These data included PSA concentrations, sociodemographic data, laboratory data, medical examination–personal life history, dietary, and comorbidities data for the second analysis. According to the following exclusion criteria, participants were left out of our study as follows: (1) female participants (*n* = 20,785); (2) aged < 40 years (*n* = 13,231); (3) participant diagnosed with prostate cancer (*n* = 314); (4) factors affecting PSA concentrations: diagnosed with prostatitis, stain drug user, received prostate biopsy within one week and had urinary system surgery within one month (*n* = 76); (5) missing PSA (734); and (6) missing serum triglycerides (*n* = 3106) [22]. There were 2910 participants out of 41,156 participants who fit into our study after conducting the screening (Figure 1). Furthermore, our study complied with the Helsinki Declaration of the World Medical Association in the process of study design and conduction. In our study, we used the data analysis based on NHANES.

In our study, serum triglyceride was the target-independent variable downloaded from the NHANES website. Collaborative Laboratory Services periodically refines these laboratory methods. More detailed information on the test principles and clinical relevance can be found on the NHANES website and articles elsewhere [23,24]. PSA concentration (ng/mL) was set as the dependent variable. Total prostate-specific antigen (PSA) from the participants’ serum were recorded by Beckman Access at the Department of Laboratory Medicine Immunology Division, and the Hybritech PSA method was used to record serum total PSA concentration (ng/mL) (https://wwwn.cdc.gov/nchs/data/nhanes/2003–2004/labmethods/l11psa_c_met_total_psa.pdf, accessed on 7 October 2021). In this study, we used total serum PSA as the result variable connected to serum triglycerides and we selected the following covariates, based on the previous articles regarding the connection to triglycerides, PSA, or prostate cancer and other covariates [25,26,27,28,29,30]. Covariates contained the sociodemographic data, laboratory data, medical examination–personal life history, dietary, and comorbidities data. Continuous variables included age (year), poverty income ratio (PIR), vitamin D (ng/mL), LDL-cholesterol (mg/dL), HDL-cholesterol (mg/dL), glycohemoglobin (%), C-reactive protein (mg/dL), BMI (kg/m^2^), alcohol (gm), and triglycerides (mg/dL). Categorical variables contained race/ethnicity, education, marital status, physical activity (MET-based rank), smoked at least 100 cigarettes in life, hypertension history, coronary heart disease history, diabetes history, and stroke. More information about the details of variables is available on the NHANES official website.

### 2.3. Statistical Analysis

Following the criteria of the CDC guidelines, we conducted a statistical analysis between the serum triglycerides and PSA level (https://www.cdc.gov/nchs/nhanes/index.htm, accessed on 7 October 2021). Serum triglyceride was expressed as the mean ± standard deviation as normally distributed. PSA concentrations and other continuous variables were expressed as the mean ± standard deviation (normal distribution). The categorical variables were expressed in percentage or frequency. Our aim was to research the potential relationship of particular participants between the serum triglycerides and PSA concentrations. First, we divided the continuous variables of serum triglycerides into four quartile concentrations. The weighted chi-square was used to calculate the categorical variable, and the weighted linear regression model was used to calculate the continuous variable between the quartile arrays which shown in the Table 1. Second, we constructed three weighted univariate and multiple linear regression models including a non-adjusted model, a minimally adjusted model, and a fully adjusted model, which are shown in Table 2 to figure out the linear relationship between the serum triglycerides and PSA concentrations. Third, subgroup analyses were performed to identify the stratified associations between triglyceride and PSA using stratified multivariate logistic regression. Furthermore, in the model-development phase, we constructed the XGBoost algorithm model to predict the relative importance of selected variables. We performed the XGBoost model to analyze the contribution (gain) of each variable to PSA concentration [31,32]. Finally, the penalty spline method constructed a fully adjusted model with a smooth curve fit to explore the potential linear relationship between the serum triglycerides and PSA concentrations. In order to prevent the bias caused by missing data, we curated the NHANES database to improve the accuracy of the analysis by using the MICE package to account for missing data [33]. The results showed that the original data represented no significant difference with the complete data. All in all, univariate and multiple analysis results were based on the calculated dataset as well as Rubin’s rules. All kinds of statistical analyses of analysis were performed by R software (Version 4.0.2) using the R package (http://www.R-project.org, The R Foundation, accessed on 7 October 2021) [34]. The software EmpowerStats also provided significant help in the process of analysis (http://www.empowerstats.com, X&Y Solutions, Inc., Boston, MA, USA). In our study, a *p*-value < 0.05 was considered statistically significant.

## 3. Results

### 3.1. Baseline Characteristics of Selected Participants Subsection

Weighted distribution of baseline characteristics is shown in Table 1 including sociodemographic data, laboratory data, medical examination–personal life history, dietary, and comorbidities data of chosen participants selected from the NHANES (2013–2010) survey. In this study, the average age of the chosen participants was 59.7 years (±12.7). Then, we divided different serum triglycerides into four quartiles (Q1–Q4). The distribution of poverty to income ratio, marital status, C-reactive protein, coronary heart disease history, diabetes history, and stroke history in Q1–Q4 of serum triglycerides showed no statistical difference with approximate similarities (*p* values > 0.05). Compared with the different groups in Table 1, the distribution of serum triglycerides showed an age difference, where younger participants had higher serum triglycerides than older ones; had more elevated LDL-C, higher glycohemoglobin, higher body mass index (BMI), more likely to have a lower education level, higher incidence of hypertension, and more likely smoked at least 100 cigarettes in life. On the other hand, participants with more elevated serum triglycerides had lower PSA concentrations, lower vitamin D, lower HDL-C, lower physical activity, and were more likely to drink less alcohol. In our study, non-Hispanic Whites were the main participants.

### 3.2. The Connection between PSA Concentrations and Serum Triglycerides

The results of the univariate and multivariate analyses by the weighted linear model are shown in Table 2. In the non-adjusted model, which adjusts for none, the PSA concentrations were reduced by 0.0014 (−0.0023, −0.0005) for each additional unit of serum triglyceride with *p* for a trend less than 0.05. In the minimally adjusted model, which adjusts for race/ethnicity, education level, poverty income ratio, and marital status, the PSA concentrations were reduced by 0.0013 (−0.0022, −0.0004) for each additional unit of serum triglyceride with *p* for the trend <0.05. The fully adjusted model that adjusts for race/ethnicity, education level, poverty income ratio, marital status, VITD, LDL-C, total cholesterol, C-reactive protein, glycohemoglobin (%), physical activity (MET-based rank) (%), BMI (kg/m^2^), smoked at least 100 cigarettes in life, drinking alcohol (gm) first day, coronary heart disease, and stroke indicated that the PSA concentrations were reduced by 0.0043 ng/mL for each additional unit of serum triglyceride (mg/dL).

### 3.3. Stratified Associations between PSA Concentrations and Serum Triglycerides

As shown in Table 3, we conducted stratified analysis by age, race, education, marital status, body mass index (BMI), and ratio of family income to assess the associations between triglycerides and PSA. It is likely that Non-Hispanic Whites, education level higher than high school, married status, and high group of ratios of family income had lower PSA concentrations, with increasing serum triglycerides displaying a significant trend (*p* for trend = 0.0036; *p* for trend = 0.0012; *p* for trend = 0.0004; and *p* for trend = 0.0002). In addition, we detected a significant difference by *p* for the interaction analyses. Variables of age and triglycerides may have an interaction effect associated with PSA concentrations (*p* for interaction < 0.0001).

### 3.4. Machine Learning Using the XGBoost Algorithm Model

In the phase of model-development and validation, we used the machine learning of the XGBoost model to determine the relative importance of selected variables associated with the PSA. Variables included triglyceride, age, body mass index (BMI), LDL-cholesterol, total-cholesterol, C-relative protein, glycohemoglobin, vitamin-D, and alcohol. According to the results of each variables’ contribution by the XGBoost model, triglyceride, age, body mass index (BMI), LDL-cholesterol, and total-cholesterol were the top five most important variables of the dataset (Figure 2). Triglyceride, as the most relative variable, was included to construct smooth curve models in our study.

### 3.5. Identification of Sensitivity Analysis

We conducted sensitivity analysis to confirm the accuracy and stability of the results. First, we converted the serum triglyceride as a continuous variable to the categorical variable in the quartile value, and then the *p*-value was calculated for trend. Surprisingly, the result of the categorical variable was consistent with the effect of the serum triglyceride as a continuous variable. To investigate the possible linear relationship between the serum triglycerides and PSA concentrations, we constructed a smooth curve based on the fully adjusted model. The relationship between the serum triglycerides and PSA concentrations was linear after adjusting for other covariates based on the fully adjusted model strategy (Figure 3). The results showed that if the serum triglyceride with each additional unit was mg/dL, the PSA concentrations were reduced by 0.0043 ng/mL. These results indicated a negative correlation between serum triglycerides and PSA concentrations.

## 4. Discussion

Our study is one of the most extensive cross-sectional studies to explore the potential association between the serum triglyceride and prostate-specific antigen (PSA) connections and is also the first to investigate this relationship in males without a history of malignant cancer in the United States based on the NHANES database. Up to now, there are no previous epidemiological studies that have reported on the association between triglyceride and PSA levels. Thus, after adjusting for the sociodemographic, laboratory, medical examination–personal life history, dietary, and comorbidities data, the serum triglyceride was negatively correlated with PSA. To avoid PSA detection bias in the process of diagnosis disease related to the prostate, it is necessary to understand that serum lipid may cause individual differences in PSA concentration and triglyceride is one aspect that needs further attention. Our study population was 2910 participants selected from the National Health and Nutrition Examination Survey (2003–2012). Machine learning of the XGBoost model was used to determine the relative importance of selected variables associated with the PSA, and triglyceride was the most relative variable. Our result shows that the PSA concentrations were reduced by 0.0043 ng/mL for each additional unit of serum triglyceride (mg/dL) and the result showed a statistical difference, which means that if 100 (mg/dL) of triglyceride is added, the PSA concentration will reduce by 4.3 ng/mL. Now, more and more of the population have dyslipidemia, which may aggravate PSA detection bias. Triglyceride, as the most relative variable, was included to construct smooth curve models to confirm the robustness of the result.

Previously, some prospective studies have provided evidence showing a potential role for lipid metabolism in PCa development, which suggests that a higher serum total cholesterol may relate to a higher grade or more aggressive prostate cancer [18,35,36,37,38,39]. The relationship between the triglyceride and the prognostic outcomes of prostate cancer have been seldomly reported and are unclear. One study showed evidence that triglycerides may influence the aggressiveness and severity of prostate cancer [25]. However, one study considered triglycerides as a protective factor because of the higher triglyceride detection in a normal population [20,21]. This result was also reported by a study from a Chinese investigation that revealed the inverse association between serum triglycerides and PSA levels [40]. Research is still needed to assess the relationship between the serum lipids and PSA level as most studies involve the population of relatively low-risk Asian men. Thus, we hypothesized that triglycerides may influence the PSA concentrations and cause detection bias, leading to this conflict in explanation. Additionally, we need to conduct further cohort experiments to understand the function of triglyceride as a protective or dangerous factor in the process of prostate cancer

The possible explanation of the inverse function of triglycerides with the PSA concentrations may be due to the detection bias between the process of prostate cancer, as shown in several recent studies [41,42,43], requires consideration. As we observed, higher concentrations of triglyceride populations are associated with lower PSA (Figure 3), so the possibility of detecting asymptomatic prostate cancer might be lower among high triglycerides. Meanwhile, if prostate cancer is already present, people with higher triglycerides may be more likely to be diagnosed at an advanced stage by PSA testing. Thus, our results considered that detection bias due to negative association between triglycerides and PSA does not explain the inverse association between the serum triglycerides and advanced prostate cancer than previously reported.

Our results support a negative association between triglycerides and PSA that may lead to detection bias, which can have implications for prostate cancer screening. Because triglycerides preferentially decrease PSA concentration in males without prostate cancer, the specificity of the PSA test for prostate cancer screening might be improved in males with high triglycerides. Thus, it is necessary to adjust the PSA threshold for further examination with various triglycerides if triglycerides can decrease PSA production by prostate tumor or change the ability of tumor-derived PSA to enter thee serum. Further studies are needed to explore the mechanism by which triglycerides influence PSA concentration and the influence on prostate cancer screening. Moreover, prospective cohort studies are still needed to confirm the causality, and serum triglyceride is involved in the occurrence and development of PCa, which needs to be verified via in vitro and in vivo experiments.

This study had several advantages compared with previously published articles. First, a large sample with a total of 2910 participants was utilized in our research. Second, machine learning of the XGBoost algorithm model was used to assess each variable’s contribution to PSA. Then, we performed a sensitivity analysis that considered and evaluated the impact of other factors, which may influence the result. Finally, a smooth curve was constructed based on the fully adjusted model to investigate the possible linear relationship between the serum triglycerides and PSA concentrations. Nevertheless, our study had some limitations in interpreting the results, which need to be considered. First, in our study, it was hard to distinguish the causality because of the intrinsic limitations of the NHANES database as a cross-sectional survey. Therefore, prospective cohort studies are still needed to confirm the causality. Additionally, we excluded participants diagnosed with prostate cancer, with factors affecting PSA concentrations and missing data. Thus, our results cannot explain the aforementioned populations. Finally, our investigation was based on the NHANES database, which is limited to the American people. Therefore, generalizability is geographically restricted. All of the above points require further evaluation and investigation in the future.

## 5. Conclusions

This nationally representative study showed that serum triglyceride is independently and negatively correlated with PSA among adult American males without a history of tumors. People with higher triglycerides would be more likely to be diagnosed with prostate cancer at an advanced stage in the future. This detection bias is unlikely to explain the inverse association between triglycerides and aggressive prostate cancer.

## Figures and Tables

**Figure 1 nutrients-14-01325-f001:**
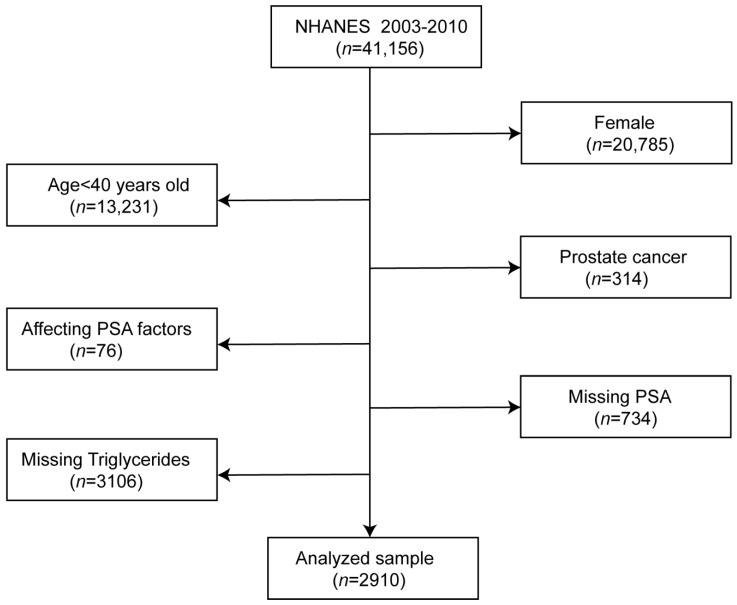
Flowchart in selecting the studying participants.

**Figure 2 nutrients-14-01325-f002:**
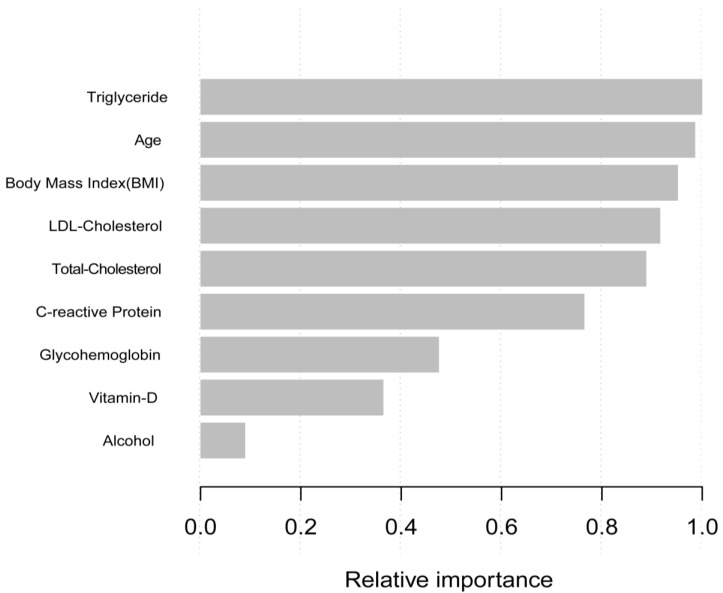
Relative importance of the selected variables using XGBoost and the corresponding variable importance score. *X*-axis indicates the importance score, which is the relative number of a variable that is used to distribute the data, *Y*-axis indicates the selected variable.

**Figure 3 nutrients-14-01325-f003:**
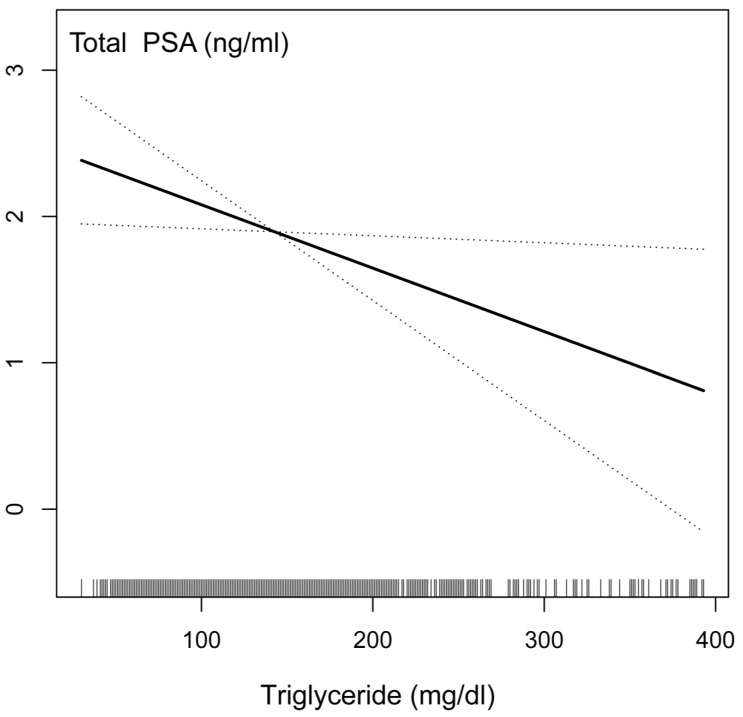
The relationship between serum triglyceride and prostate-specific antigen (PSA) connections.

**Table 1 nutrients-14-01325-t001:** Baseline characteristics of the selected participants.

Triglycerides (mg/dL)	Q1	Q2	Q3	Q4	*p*-Value
N	718	719	744	729	
PSA ng/ml	1.56 ± 2.60	1.74 ± 3.11	1.40 ± 1.86	1.30 ± 1.62	0.0023
Sociodemographic variables					
Age, mean ± SD (years)	56.16 ± 11.94	55.84 ± 11.31	56.46 ± 11.36	54.48 ± 10.70	0.0041
Poverty to income ratio, mean ± SD (years)	3.31 ± 1.57	3.40 ± 1.55	3.32 ± 1.56	3.30 ± 1.61	0.6691
Race/ethnicity (%)					<0.0001
Mexican American	3.91	5.39	6.57	8.30	
Other Hispanic	2.47	2.69	3.72	4.16	
Non-Hispanic White	76.52	75.82	78.13	75.67	
Non-Hispanic Black	13.07	10.12	5.56	5.38	
Other race/ethnicity	4.02	5.98	6.02	6.50	
Education (%)					0.0335
Less than high school	19.27	16.60	21.00	20.20	
High school	22.65	23.99	22.76	27.83	
More than high school	58.08	59.41	56.24	51.97	
Marital status (%)					0.2121
Married	71.14	75.20	69.96	73.24	
Single	24.19	19.50	25.35	21.98	
Living with a partner	4.67	5.30	4.69	4.77	
Variables of laboratory data					
VITD, mean ± SD (ng/mL)	68.95 ± 21.51	60.86 ± 19.91	64.92 ± 19.99	60.63 ± 19.22	<0.0001
LDL-C, mean ± SD (mg/dL)	112.74 ± 32.98	121.48 ± 31.45	123.05 ± 36.68	119.55 ± 36.91	<0.0001
HDL-C, mean ± SD (mg/dL)	58.95 ± 15.70	51.76 ± 13.34	45.75 ± 9.90	40.82 ± 9.75	<0.0001
Glycohemoglobin (%)	5.59 ± 0.80	5.68 ± 0.87	5.70 ± 0.89	5.90 ± 1.27	<0.0001
C-reactive protein, mean ± SD (mg/dL)	0.37 ± 0.97	0.38 ± 1.07	0.47 ± 1.36	0.35 ± 0.40	0.0743
Medical examination and personal life history					
Physical activity (MET-based rank) (%)					
Sits					0.0013
Walks	21.14	18.46	25.42	19.91	
Light loads	41.71	50.94	50.31	51.50	
Heavy work	22.80	24.21	15.82	20.17	
Body mass index, mean ± SD (Kg/m^2^)	14.35	6.40	8.45	8.41	
Smoked at least 100 cigarettes in life	27.38 ± 5.39	28.46 ± 6.47	29.58 ± 5.59	30.60 ± 5.17	<0.0001
Yes					0.0253
No	54.13	58.29	59.17	61.96	
Dietary interview-individual foods	45.87	41.71	40.83	38.04	
Alcohol, mean ± SD (gm)					
**Comorbidities** (%)	19.45 ± 36.73	14.22 ± 34.31	13.77 ± 29.79	15.30 ± 34.20	0.0079
Hypertension history					
Yes					0.0245
No	35.25	36.41	46.94	46.49	
Coronary heart disease	64.75	63.59	53.06	53.51	
Yes					0.1771
No	8.08	6.56	9.71	7.97	
Diabetes history	91.92	93.44	90.29	92.03	
Yes					0.0629
No	9.35	11.41	13.16	13.39	
Borderline	88.81	86.27	84.55	83.16	
Stroke	1.84	2.31	2.29	3.45	
Yes					0.4934
No	2.86	3.49	4.22	4.12	
	97.14	96.51	95.78	95.88	

Q1–Q4: Grouped by quartile according to the serum triglycerides. Our data included PSA concentrations, sociodemographic data, laboratory data, medical examination–personal life history, dietary, and comorbidities data for the second analysis.

**Table 2 nutrients-14-01325-t002:** Univariate and multivariate analyses by the weighted linear model.

Exposure	Non-Adjusted Model	Minimally Adjusted Model	Fully Adjusted Model
Triglyceride	−0.0014 (−0.0023, −0.0005), 0.001309	−0.0013 (−0.0022, −0.0004), 0.003832	−0.0043 (−0.0082, −0.0005), 0.027856
Triglyceride			
Q1 Q2 Q3 Q4	Ref 0.1045 (−0.2349,0.4439), 0.546189 −0.3022 (−0.6387,0.0343), 0.078467 −0.4598 (−0.7980, −0.1216) 0.007755	Ref 0.0684 (−0.2852, 0.4220) 0.704653 −0.2621 (−0.6169, 0.0927) 0.147739 −0.4501 (−0.8093, −0.0909) 0.014117	Ref 0.2846 (−0.3559, 0.9250) 0.384057 −0.4040 (−1.0497, 0.2416) 0.220247 −0.5155 (−1.2396, 0.2085) 0.163151
*p* for trend	<0.001	0.002	0.049

Non-adjusted model adjusts for none. Minimally adjusted model adjusts for race/ethnicity, education level, poverty income ration, and marital status. Fully adjusted model adjusts for race/ethnicity, education level, poverty income ration, marital status, VITD, LDL-C, total cholesterol, C-reactive protein, glycohemoglobin (%), BMI (kg/m^2^), physical activity (MET-based rank) (%), smoked at least 100 cigarettes in life, drinking alcohol (gm) first day, coronary heart disease, and stroke.

**Table 3 nutrients-14-01325-t003:** Effect size of triglycerides on PSA in the prespecified and exploratory subgroup.

Triglycerides (mg/dL)	N	β	95% CI	*p*-Value	*p* for Interaction
Stratified by age					<0.0001
<60	1475	−0.0012	(−0.0028, 0.0003)	0.1241	
60–80	1153	−0.0038	(−0.0090, 0.0014)	0.1557	
>80	282	−0.0225	(−0.0472, 0.0023)	0.078	
Stratified by race					0.3315
Mexican American	496	−0.0012	(−0.0054, 0.0031)	0.5987	
Other Hispanic	213	0.0017	(−0.0061, 0.0094)	0.678	
Non-Hispanic White	1585	−0.006	(−0.0100, −0.0020)	0.0036	
Non-Hispanic Black	498	−0.0125	(−0.0281, 0.0031)	0.1184	
Other race/ethnicity	118	−0.0092	(−0.0228, 0.0044)	0.1946	
Stratified by education					0.1640
Less than high school	935	−0.0082	(−0.0173, 0.0009)	0.0791	
High school	669	−0.0055	(−0.0124, 0.0014)	0.1168	
More than high school	1306	−0.0059	(−0.0094, −0.0024)	0.0012	
Stratified by marital status					0.8274
Married	1981	−0.0073	(−0.0113, −0.0033)	0.0004	
Single	789	−0.0038	(−0.0123, 0.0047)	0.3763	
Living with a partner	136	−0.0064	(−0.0126, −0.0001)	0.0527	
Stratified by BMI					0.1168
<25	710	−0.0092	(−0.0184, −0.0000)	0.0504	
25–28	718	−0.0052	(−0.0172, 0.0068)	0.3964	
>28	1424	−0.0029	(−0.0056, −0.0003)	0.0305	
Stratified by ratio of family income					0.5872
Low group	896	−0.0038	(−0.0081, 0.0005)	0.0873	
Median group	898	−0.0066	(−0.0157, 0.0026)	0.1594	
High group	906	−0.0093	(−0.0142, −0.0044)	0.0002	

Note 1: Above adjusts for race/ethnicity, education level, poverty income ratio, marital status, VITD, LDL-C, total cholesterol, C-reactive protein, glycohemoglobin (%), BMI (kg/m^2^), physical activity (MET-based rank) (%), smoked at least 100 cigarettes in life, drinking alcohol (gm) first day, coronary heart disease, and stroke. Note 2: In each case, the model was not adjusted for the stratification variable itself.

## Data Availability

All data are available at NHANES website https://www.cdc.gov/nchs/nhanes/index.htm (accessed on 7 October 2021).

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
