# Peer review of "Association between Serum Triglycerides and Prostate Specific Antigen (PSA) among U.S. Males: National Health and Nutrition Examination Survey (NHANES), 2003–2010"

_nutrients, 2022, doi:10.3390/nu14071325_

Round 1
Reviewer 1 Report
I read with extremely interest the present article. I have some minor comments:
- Would it be possible include data on histology for prostate cancer?
- Do you have any data free testosterone?
Reviewer 2 Report
Although the hypothesis that lipid and glucose metabolism 53 play a critical role in prostate cancer has been questioned by several studies so far, the manuscript presented does not demonstrate that there is a possible role in prostate cancer development, but rather that there is a statistical significance of data gathered in a nationwide database.
The manuscript is difficult to read due to the enumeration of statistical results. Although needed, these statistics should be simplified in order to increase readability.
Multiple redundancies and repetition throughout the manuscript. For example:
Page 6 line 177-178: the PSA 177 concentrations are reduced by 0.0043 (-0.0082, -0.0005) with P for trend less than 0.05
Page 8 line 223: The PSA concentrations are reduced by 0.0043 ng/mL (95% CI: -0.0082, -223 0.0005), followed by the p for trend less than 0.05.
Page 10 line 243: the PSA concentrations are reduced by 0.0043 ng/mL (95% CI: -0.0082, -0.0005), 243 following with the P for trend less than 0.05.
Revision of English grammar is needed.
Examples:
Page 4 line 157: “participants with higher serum triglycerides were younger than older”
Page 9 line 236: “make PSA become a better tool to diagnosis the disease”
The authors should discuss the clinical implications of reduction of PSA concentrations for each additional unit of serum triglyceride. From a clinical point of view, reduction of 0.0014 ng from 4 ng is still 4. Smaller, yes, but still 4. Also, the majority of PSA test provide only 2 decimals so a reduction from 1.4*10-4 from 4.11 is 4.11.
Page 10 line 251: “Some experimental studies also 251 showed the triglyceride-rich remnant could upregulate signaling like the MEK/ERK and 252 Akt pathways to promote carcinogenesis in the process of cell growth, cycle arrest, lipid 253 metabolism and apoptosis by using in vitro models”
This phrase has no relation with the scope of this article.
Round 2
Reviewer 2 Report
The authors have made the recommended modifications to the manuscript and made it more readable.
No further comments from my part